# Detection of *Methanobrevobacter smithii* and *Methanobrevibacter oralis* in Lower Respiratory Tract Microbiota

**DOI:** 10.3390/microorganisms8121866

**Published:** 2020-11-26

**Authors:** Yasmine Hassani, Fabienne Brégeon, Gérard Aboudharam, Michel Drancourt, Ghiles Grine

**Affiliations:** 1IHU Méditerranée Infection, 13005 Marseille, France; yasmine_hassani@yahoo.fr; 2IHU Méditerranée Infection, Aix-Marseille Université, IRD, MEPHI, 13005 Marseille, France; Fabienne.BREGEON@ap-hm.fr (F.B.); gerard.aboudharam@univ-amu.fr (G.A.); michel.drancourt@univ-amu.fr (M.D.); 3Faculté d’odontologie, Aix-Marseille-Université, 13005 Marseille, France

**Keywords:** methanogens, *Methanobrevibacter smithii*, *Methanobrevibacter oralis*, respiratory tract, microbiota

## Abstract

Methanogens, the sole microbes producing methane, are archaea commonly found in human anaerobic microbiota. Methanogens are emerging as opportunistic pathogens associated with dysbiosis and are also detected and cultured in anaerobic abscesses. Their presence in the respiratory tract is yet unknown. As a preliminary answer, prospective investigation of 908 respiratory tract samples using polyphasic approach combining PCR-sequencing, real-time PCR, fluorescent in situ hybridization (FISH), and methanogens culture was carried out. *Methanobrevibacter smithii* and *Methanobrevibacter oralis* DNA sequences, were detected in 21/527 (3.9%) sputum samples, 2/188 (1.06%) bronchoalveolar lavages, and none of 193 tracheo-bronchial aspirations. Further, fluorescence in situ hybridization detected methanogens in three sputum investigated specimens with stick morphology suggesting *M. oralis* and in another one bronchoalveolar lavage sample investigated, diplococal morphology suggesting *M. smithii*. These observations extend the known territory of methanogens to the respiratory tract and lay the foundations for further interpretation of their detection as pathogens in any future cases of isolation from bronchoalveolar lavages and the lungs.

## 1. Introduction

Methanogenic archaea (referred to in this study as methanogens), the sole organisms known to produce methane, have been characterized in the oral cavity microbiota (*Methanobrevibacter oralis, Methanosarcina mazei*, *Methanobacterium congolense*, *Methanobrevibacter smithii*, and *Methanobrevibacter massiliense*) [1,2,3], in the gastric microbiota in newborns colonized by *M. smithii* [4], in digestive tract microbiota (*M. smithii, M. oralis, Methanosphaera stadtmanae*, *Methanomassilicoccus luminyensis*, and *Methanobrevibacter arboriphilicus*) [5,6] and recently, in urinary tract microbiota [7] (Figure 1). *M. smithii* and *M. oralis* have been associated with dysbiosis affecting the oral cavity, the gut [8,9,10], and the vaginal cavity in the case of vaginosis [11]. Furthermore, these two methanogens have been detected and isolated from anaerobes collected from abscesses including life-threatening brain abscesses [12,13], sinusal abscesses [14], and abscesses in other locations [8,9,12,13,15]. Methanogens have seldom been detected in the respiratory tract [14]; here, we prospectively investigated a series of respiratory tract samples for the polyphasic detection of methanogens to find out more about their presence in this poorly explored territory.

More specifically, we report on the medical observations of two unrelated patients in whom methanogens were detected in the lung, but not in the corresponding upper respiratory tract samples.

## 2. Materials and Methods

### 2.1. Clinical Samples

This prospective study included all the respiratory tract samples including sputum, bronchial aspirates, and bronchoalveolar lavage samples (Figure 1), collected between February and April 2019 at the diagnostic laboratory of IHU Méditerranée Infection as part of its routine activities after the approval of the IHU Méditerranée Infection Ethics Committee in date of 29/09/20 with ethic approval code n° 2016-011. Briefly, the specimens were analyzed accordingly: in nonintubated patients, sputum was collected after two to three cough efforts to favor expectoration followed by direct deposition of the sample in a plastic container with screw top cover, and then saliva specimens were discarded after examination as previously reported [16]. In intubated patients, the bronchial aspirates were obtained using a sterile suction catheter connected to a sterile mucus collector after gentle introduction in the endotracheal tube and light aspiration. Bronchoalveolar lavages were performed through a fiberoptic bronchoscope by instilling up to four portions of 50 mL of isotonic saline solution into the lung segment corresponding to the radiologically most abnormal region, followed by gentle aspiration with the same 50 mL syringe for lavage fluid recovery. All samples were transferred to the laboratory for analysis.

All the samples were transferred to the laboratory and immediately placed into a Hungate tube containing 5 mL of a specific transport medium, as previously described [17,18,19].

### 2.2. Molecular Detection of Methanogens

All samples were examined by standard polymerase chain reaction (PCR) assays targeting the 16S rRNA gene in an automated 5A PTC-200 thermal cycler (MJ Research, Waltham, MA, USA). The 16S gene was amplified using broad-range rRNA primers SDArch0333aS15 (5′-TCCAGGCCCTACGGG-3′) and SDArch0958aA19 (5′-YCCGGCGTTGTTGAMTCCAATT-3′) (Eurogentec, Seraing, Belgium) [4]. The PCR reaction was performed as previously described [4]. ChromaPro (http://technelysium.com.au/wp/chromaspro/) and blast programme of NCBI (https://blast.ncbi.nlm.nih.gov/Blast.cgi) were used for sequence analyses. A total of 5μL of distilled water were used as a negative control for sputum samples and bronchial aspirates, and 5μL of 0.9% NaCl saline for bronchoalveolar lavage.

The PCR products were sequenced using the same primers as used for PCRs following this program: a 1-min denaturation step at 96 °C, followed by 25 cycles of denaturation of 10 s at 96 °C, a 20-s annealing at 50 °C and a 4-min extension at 60 °C. The MultiScreen 96-well plates Millipore (Merck, Molsheim, France) containing 5% of Sephadex G-50 (Sigma-Aldrich, Saint-quentin-fallavier, France) were used to purify the sequencing products. The sequences were analyzed and edited following the protocol described previously [18]. Further, confirmatory real-time PCR was performed using primers and probes specific to *M. oralis* cnp602F primer 5-GCTGGTGTAATCGAACCTAAACG-3 (Eurogentec); cnp602R primer 5-CACCCATACCCGGATCCATA-3 (Eurogentec) and the probe cnp602-FAM 5-AGCAGTGCACCTGCTGATATGGAAGG-3 (Eurogentec) [9] and *M. smithii* (Smit.16S-740F, 5-CCGGGTATCTAATCCGGTTC-3 (Eurogentec); Smit.16S-862R, 5-CTCCCAGGGTAGAGGTGAAA-3 (Eurogentec) and the probe Smit.16S FAM, 5-CCGTCAGAATCGTTCCAGTCAG-3 (Eurogentec) [19]) species. Amplification occurred in a reaction with a volume of 20 μL including 15μL of mix and 5μL of DNA extract. The amplification reaction was carried out in a Roche real-time PCR system (LightCycler 480 II) and BIO-RAD (CFX96TM Real-Time System), using the following protocol: one cycle at 95 °C for 5 min, followed by 40 cycles at 95 °C for 1 s and 60 °C for 35 s and finally 45 °C for 30 s. A total of 5 μL of distilled water were used as a negative control for sputum samples and bronchial aspirations, and 5 μL of 0.9% NaCl saline for bronchoalveolar lavage.

### 2.3. Isolation and Culture

A volume of 250 μL of each sample diluted in PBS was seeded under anaerobic chamber in a sterile Hungate tube (Dominique Dutscher, Brumath, France) [20,21]. Each Hungate tube contained 5 mL of SAB broth [22] supplemented with glutathione (0.1 g/L) (Sigma-Aldrich, Saint-Quentin-Fallavier, France), ascorbic acid (1 g/L; VWR International, Leuven, Belgium), and uric acid (0.1 g/L) (Sigma-Aldrich). The pH was adjusted to 7.5 with NaOH (10 M) and 5 mL of SAB broth were inoculated with *Bacteroides thetaiotaomicron* (104 cells/mL) for hydrogen production [23] and the Hungate tube containing the sample and *B. thetaiotaomicron* in SAB broth were incubated at 37 °C for 7 days. Subculture was seeded on a Petri dish containing SAB medium supplemented with 15 g/L agar (Sigma-Aldrich) and deposited in the upper chamber of a double-chamber box. Methanogen colonies appeared after 9–12 days of incubation [23].

### 2.4. Fluorescent In Situ Hybridization

Only PCR positive samples were analyzed by FISH using the method previously described [1,4]. In brief, 400 μL of the sample were washed twice with PBS. The final pellet was suspended in 400 μL PBS and fixed by adding 150 μL of 4% paraformaldehyde and incubated overnight at 4 °C and then the hybridization was carried out using the archaea-specific Arch915 probe (GTGCTCCC CCGCCAATTCCT) [24] staining the archaeal 16S rRNA and Eub 488 probe staining the bacterial 16S rRNA gene (TTCATTGCRTAGTTWGGRTAGTT) previously described by Luton et al. [25]. Once the washes were done and the hybridization of 16 h completed, the slides were observed using the confocal microscope LSM800 with different wavelengths, including 488FITC nm for reading the archaeal 16S rRNA probe, 545RHOD nm for reading the bacterial 16S rRNA probe, and 647Cy5nm for reading the nonspecific probe and DAPI.

We used a negative methanogen PCR sputum sample and a negative methanogen bronchoalveolar lavage sample as negative controls.

## 3. Results

### 3.1. Molecular Detection

#### 3.1.1. PCR-Sequencing

In the presence of negative control which remained negative, PCR-sequencing disclosed methanogens in 21/527 (3.98%) of the sputum samples including 19 *M. oralis*-positive samples exhibiting 99% sequence similarity with *M. oralis* strain ZR (GenBank: NR_104878.1) and two M. *smithii*-positive samples exhibiting 99% sequence similarity with *M. smithii* ATCC 35,061 (GenBank NR: 074235.1); in 2/188 (1.06%) of bronchoalveolar lavage samples (including one *M. oralis*-positive sample exhibiting 100% similarity with *M. oralis* strain VD9 (GenBank: LN898260.1) and one *M. smithii*-positive sample exhibiting 99.97% similarity with *M. smithii* strain C2 CSUR P5816 (GenBank: LR590664.1). None of the bronchial aspirate samples were PCR-positive.

#### 3.1.2. Real-Time PCR

In the presence of negative control which remained negative, all the positive samples by PCR-sequencing were positive by real-time PCR. Real-time PCR analyses targeting the *M. smithii* 16S rRNA gene yielded a median Ct of 35.7 ± 0.47, indicating a *M. smithii* load of 1.32 × 10^4^ ± 0.56 × 10^4^/mL in the case of two sputum positive samples and Ct of 36.2 indicative of an *M. smithii* load of 0.8 × 10^4^/mL in case of one bronchoalveolar lavage positive sample.

Real-time PCR analyses targeting the *M. oralis* cnp-60 gene yielded a median Ct of 33.2 ± 1.07 indicative of an *M. oralis* load of 2.61 × 10^5^ ± 1.23 × 10^5^/mL in the case of 19 sputum positive samples and Ct of 35.3 indicative of an *M. oralis* load of 1.41 × 10^4^ in case of one bronchoalveolar lavage positive sample (Table 1).

#### 3.1.3. Phylogenetic Analysis

All the positive sequences obtained cluster with the two methanogen species *M. smithii* and *M. oralis* (Figure 2).

### 3.2. Isolation and Culture

All the samples analyzed were negative for methanogen culture.

### 3.3. Fluorescent In Situ Hybridization

Of the 19 sputum samples positive for *M. oralis* by PCR, three were FISH positive and of the two samples positive for *M. smithii* by PCR none were FISH positive.

In the case of bronchoalveolar lavage sample, we obtained only one positive sample by FISH out of the two samples analyzed, which was positive for *M. smithii* by PCR (Figure 1).

FISH investigation yielded microorganisms presenting a diplococcus morphology characteristic for *M. smithii* for the one bronchoalveolar lavage sample and stick morphology characteristic for *M. oralis* for the three sputum positive samples (Figure 3).

## 4. Discussion

The observations here reported, pioneer the detection of the methanogens *M. oralis* and *M. smithii* in sputum and bronchoalveolar fluid. These observations are technically validated by the negativity of the negative controls introduced in each experimental batch. However, positive detections must be interpreted with caution as they could result from the contamination of the respiratory tract samples by the oral cavity fluid microbiota, already known to harbor the very same methanogens [1,10]. In this report, we did not detect methanogens in any of bronchial aspirate samples whereas methanogen detection was positive in 21 sputum samples and two bronchoalveolar samples, all devoid of evidence for oral fluid contamination. The fact that the bronchial aspirate sample was negative for methanogens, further eliminate a mere contamination of the sputum and of the bronchoalveolar sample. Later observations are in agreement with previously reported detection of archaea by metagenomic in 36 bronchoalveolar lavages; although later study found a majority of archaea of the superphylum DPANN, and the Woesearchaeota members; but no methanogens [26]. This is the first time that the presence of *M. oralis* and *M. smithii* has been reported by molecular technique and FISH in sputum and bronchoalveolar fluid.

Detection of methanogens in the sputum and bronchoalveolar fluid requires clarification on the source of methanogens. While the oral cavity previously shown to culture both *M. oralis* and *M. smithii* [1,27] is one potential such source of inhaled methanogens, translocation from gut compartment could also be a mechanism of lung contamination by the methanogens. Indeed, *M. smithii* and *M. oralis* have been recovered from the gut microbiota [4,6,28,29,30] and have been previously found in different fluids including the urine [7] and milk [28]. Translocation has been advocated as the mechanism of fluid contamination [11,30]. Although saliva specimens have been discarded, nevertheless, the presence of methanogens in sputum with a prevalence of 3.9% (higher than in BAL) could be partly due to contamination through saliva which is colonized by the methanogenic Archaea, mainly *M. oralis* [1]. The role of methanogens in the lung pathology cannot be established in part because living methanogens could not be cultured, due to inappropriate storage condition of sputum and bronchoalveolar lavage samples and as methanogens are oxygen intolerant microorganisms, they are unlikely to colonize lung alveoli where oxygen is abundant. Indeed, *M. smithii* and *M. oralis* are strict anaerobic methanogens being killed by an average 10-min exposure to ambient air [31]. Accordingly, their survival in safe lung alveoli is improbable, while this may not be the case in lung disease creating anaerobiosis lung pockets.

The detection of specific pieces of DNA is not conclusive of detection of living microorganisms. Indeed, the prevalence of methanogens detected by molecular biology in human samples including oral fluid [1] and vaginal fluid [11] exceeds the prevalence detected by culture.

Noteworthy, the two methanogens’ positive bronchoalveolar lavage patients here described were tobacco-smokers and tobacco-smoking has been associated with increased prevalence of methanogens (and Gram-positive bacteria) in the saliva [1,32]. Interestingly, lung lesions increasingly reported in e-smoking have been thoroughly investigated for any opportunistic pathogen, except for methanogens [33,34,35]. We propose to search for methanogens in such lesions using appropriate methods such as those here reported as methanogens could likely be the missing links between e-smoking and vaping-associated lung injuries.

## Figures and Tables

**Figure 1 microorganisms-08-01866-f001:**
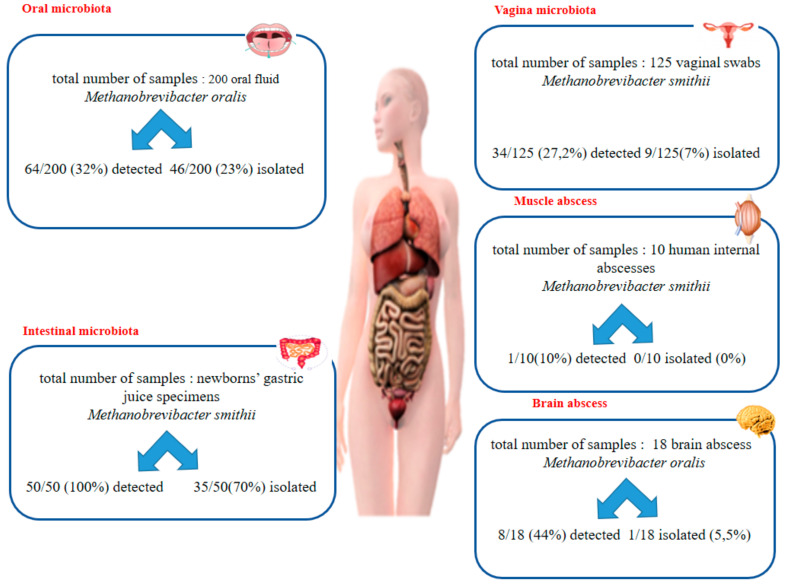
The comparative results of PCR detection and culture detection in different human microbiota.

**Figure 2 microorganisms-08-01866-f002:**
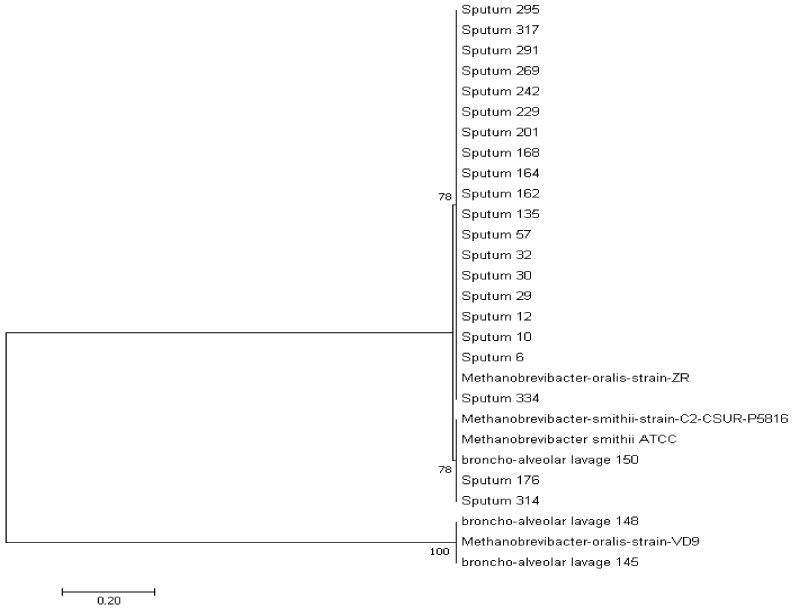
Archaeal 16S rRNA gene sequence-based phylogenetic tree illustrating the phylogenetic position of all the methanogens’ positive sequences relative to other phylogenetically close neighbors. Only bootstrap values > 65% are indicated at nodes.

**Figure 3 microorganisms-08-01866-f003:**
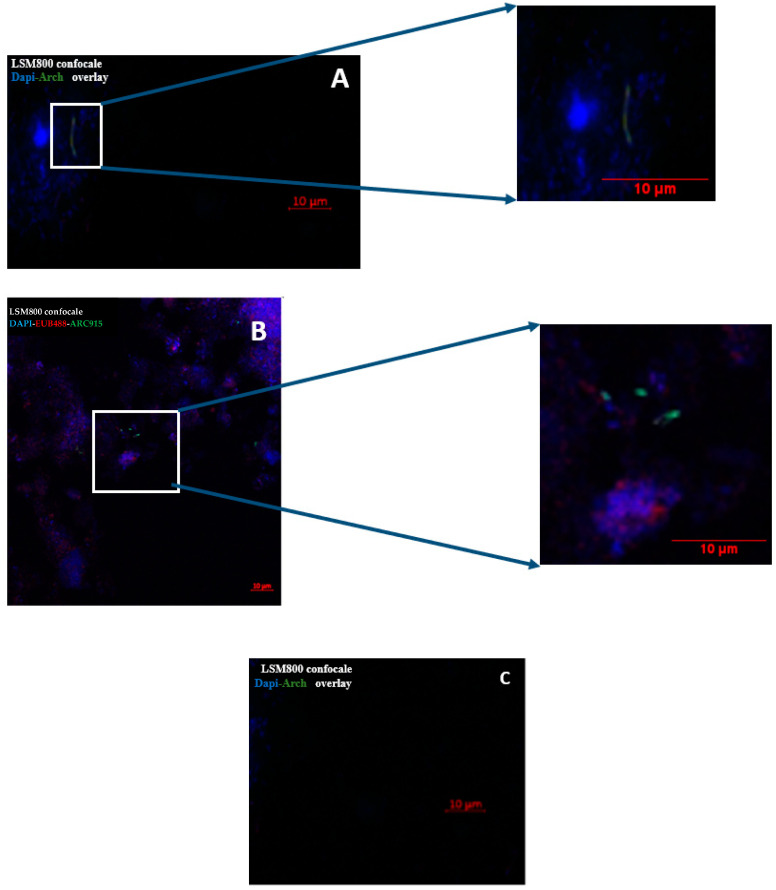
Fluorescent in situ hybridization (FISH) detection of methanogens species in (I) sputum sample and in (II) bronchoalveolar lavage. (**A**) Sputum sample positive by RT-PCR *M. oralis*; (**B**) bronchoalveolar lavage sample positive by RT-PCR *M. smithii*; (**C**) sputum sample negative by RT-PCR *M. oralis* (negative control). In blue: DAPI; in red: the probe staining the bacterial 16S rRNA; in green: the probe staining the archaeal 16S rRNA. Scale bar = 10 µm.

**Table 1 microorganisms-08-01866-t001:** Summary of the results obtained by PCR- based analysis, by culture method and by FISH.

Nature of Sample	Sputum	Bronchoalveolar Lavage	Bronchial Aspirates
Number of analyzed samples	527	188	193
Archaea 16S rRNA PCR	21/527 (3.9%)	2/188 (1.06%)	0
Sequencing	*M. oralis* (*n* = 19); *M. smithii* (*n* = 2)	*M. oralis* (*n* = 1); *M. smithii* (*n* = 1)	0
RT PCR (*M. smithii*)	*n* = 2 (with load of 1.32 × 10^4^ ± 0.56 × 10^4^/mL)	*n* = 1 (with load of 0.8 × 10^4^/mL)	0
RT PCR (*M. oralis*)	*n* = 19 (with load of 2.61 × 10^5^ ± 1.23 × 10^5^/mL)	*n* = 1 (with load of 1.41 × 10^4^/mL)	0
Methanogen culture	0	0	0
FISH	*n* = 3	*n* = 1	0

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
