# Peer review of "Detection of Methanobrevobacter smithii and Methanobrevibacter oralis in Lower Respiratory Tract Microbiota"

_microorganisms, 2020, doi:10.3390/microorganisms8121866_

Round 1

Reviewer 1 Report

Dear Authors, your manuscript is extremely sloppy. It is surprising to compare this text with a superbly designed study by your own group (DOI:10.1038/s41598-018-27372-7). Figure is uninformative and not properly described.

Reviewer 2 Report

Line 46: remove either “prospective” or “prospectively”
Lines 46 – 50: Can you please provide a brief description of how each sample type was collected?
Line 102-103: Was a positive control used?
Lines 118 and 120: Should this be 10^4 equivalents based on a standard curve?
Line 144: remove “was”
Line 147: change “precaution” to “caution”
Line 151: How was evidence of oral fluid contamination investigated?
Line 166: include storage conditions of the samples and collection methods in the M/M
Line 169: change why to while
Line 176-178: Last line seems to be a stretch based on 2 patients. I would remove.
Discussion: I would add discussion on the detection of microbial sequence vs live organisms that can be cultured. How often are methanogens cultured out of other samples? Hypotheses other than storage conditions which may have impacted your ability to culture out the methanogens?

Round 2

Reviewer 1 Report

Dear Authors!

  1. You detected only DNA of Methanogens in the sputum samples and bronchoalveolar lavages, and “living methanogens could not be cultured”. So, you can not exclude the translocation of this DNA or dead bacteria from the gastrointestinal tract. Therefore, it is at least incorrect to include in the title of the manuscript the assumption that methanogens are included in the lung microbiota. The title of the manuscript must be changed.
  2. Figure 1 is redundant. It illustrates well-known methods and can be used in an introductory lecture course or in a popular science publication, but not in a scientific article.
  3. Figure 4 fits well into the first paragraph of the introduction.It is advisable to move it and designate Figure 1.
  4. Figures 3A-3D are large, and the informative part, taken in a square, is very small.It is advisable to increase it by reducing the empty field.
  5. In the abstract you claim “More specifically, we report on the medical observations of two unrelated patients in whom methanogens were detected in the lung, but not in the corresponding upper respiratory tract samples” In fact, you mention in discussion, that the two methanogen’s positive bronchoalveolar lavage patients were tobacco-smokers, and do not indicate in the results other samples related to these patients. Is it correct to write in this case aboutt report of the medical observations?

Author Response

Thank you. 
